

# Evaluation of soil intervention values in mine tailings in northern Chile

Elizabeth Lam Esquenazi[1], Brian Keith Norambuena[2], Ítalo Montofré Bacigalupo[3] and María Gálvez Estay[1]

[1] Chemical Engineering Department, Universidad Católica del Norte, Antofagasta, Chile
[2] Department of Computing and Systems Engineering, Universidad Católica del Norte, Antofagasta, Chile
[3] Metallurgical and Mining Department, Universidad Católica del Norte, Antofagasta, Chile

Corresponding author
Elizabeth Lam Esquenazi,
elam@ucn.cl

## ABSTRACT

The aim of this work is to show a methodological proposal for the analysis of soil intervention values in mine tailings in order to determine the intervention requirements in the commune of Andacollo in northern Chile. The purpose of this analysis is to guide the intervention policies of both private and public organizations. The evaluation method is based on the Dutch legislation. The usability of the proposed methods depends on the available geochemical data from soil samples; in particular, we tackle the case when information regarding clay percentage in the soil is not available. We use the concepts of a threshold factor and an adjusted threshold factor to calculate a weighted intervention ranking. In order to illustrate the utility of this methodological proposal, a case study is carried out with the prescribed approach. In particular, this work presents an analysis of the elements of environmental significance related to the mining activity (Hg, Cd, Pb, As, Cu, Ni, Zn, Cr) in the commune of Andacollo, Coquimbo Region, Chile. The analyzed samples are used to determine where the intervention of tailing deposits is necessary and where a solution to these environmental liabilities is required as soon as possible. Out of the 81 samples evaluated, it was found that 18 require a potential intervention, and of these samples, seven of them are associated with abandoned tailings that, in some cases, are located close to the town center itself, one sample is associated with active tailings and the other 10 with inactive tailings.

## INTRODUCTION

The Rio de Janeiro Summit of 1992 marked a historic milestone in the international commitment to protecting the environment (*Sequeiros, 1998*). In this summit, the importance of soils was recognized, as well as the need to protect them and their potential uses in the context of sustainable development, in particular against the contamination caused by activities of anthropogenic origin. This has led to the development of soil quality indicators with the purpose of preserving and improving the productivity of soils (*Doran & Parkin, 1996*; *Azapagic, 2004*; *Andrews, Karlen & Cambardella, 2004*; *Römbke et al., 2016*; *De Graaf, Platjouw & Tolsma, 2017*; *Turpin et al., 2017*).

In Chile, as in many parts of the world, there is a great number of mining environmental liabilities, mainly composed of tailings, which are potential risk sources for people and the environment. The great number of tailings distributed throughout Chile, many of which are abandoned with no one in charge of them, is a big problem for the State of Chile since the application of control measures requires large amounts of money. Therefore, it is imperative to have an effective and economical tool that allows determining whether a tailing requires intervention or not.

Mining is Chile's main economic activity and the fundamental pillar of its growth. Thanks to the great geological diversity of its territory, large-scale deposits of different minerals can be found in the country. In the case of copper, Chile has one of the largest reserves worldwide (*Oyarzún & Oyarzún, 2011*), being the leading country in the production of this resource. The national production of copper, which is concentrated mainly in the northern part of the country, reached 5,764,000 tons of fine copper in 2015 and is expected to grow at an annual rate of 1.19% by 2020 (*COCHILCO, 2016*).

An important point to consider is the strong increase in the production of copper concentrates that is projected to rise by 32.1% within the next 10 years, to the detriment of the production of cathodes due to the depletion of oxidized resources and the lack of new hydrometallurgical projects (*Cifuentes, 2016*). The strong growth of the production of concentrates, added to the low grades of ores, has affected the generation of tailings from concentrator plants. For 10 years (2002–2011) over 2,150 million tons of tailings have been produced that are generally not being treated. In addition, the average generation of tailings by 2020 is projected to amount to more than 380 million tons per year (*Muñoz Aracena, 2017*). Additionally, mining causes a significant impact on the environment such as air pollution associated with the movement of land in the case of open-pit mining and emissions of toxic gases in smelters. Regarding water resources, this industry brings about their depletion and pollution due to discharges of industrial liquid waste to superficial channels, or by infiltration from tailings to underground layers. Also, ecosystems are affected by the mobilization of leached metals from tailings deposits. In addition, there may be a potential archaeological impact and an impact on the flora and fauna of the area.

Chile is a country that historically, besides copper, has focused on the mining of iron, gold, silver, lithium, iodine, and potassium, without giving space to the exploitation of new raw materials such as cobalt, antimony, niobium, and rare earth metals that have acquired increasingly important roles in modern industry and whose future supply has become a concern for the industrialized countries given the dominance of China in the market for these elements.

In the case of Chile, copper production schemes have considered the recovery of secondary elements such as gold, silver, and molybdenum. However, the rest of the elements remain as part of the waste, such as tailings, gravel, or slag. On the other hand, there are the environmental problems associated with mining tailings in Chile, totaling 696 deposits until 2016, of which 84% correspond to non-active or abandoned tailings. Of the 584 non-active and abandoned tailings deposits, 310 do not have an environmental rating resolution available for its project and 477 do not have a closure

plan because, due to their age, they have been closed and abandoned before the change in regulations (*National Geology and Mining Service of Chile (SERNAGEOMIN), 2016*). Added to the unknown state of regularization of these tailings, an important growth of these deposits that would result in 81% more mass of tailings is expected, according to the projections of *COCHILCO (2016)*. This is explained by the growing production of copper concentrates and the low-grade ores being exploited.

Despite the advances of the international community in this matter, Chile still has a pending debt due to the lack of regulations for soil quality. This is particularly harmful to the population due to the development of mining activities that bring about a series of negative impacts on the soil in several regions of the country. These mining liabilities have been the result of a historical mining that had very weak regulations regarding the closure stage. Fortunately, the Law 20,551 was promulgated in 2012, which demands that all mining sites present a closure plan prior to starting the mining project (*National Geology and Mining Service of Chile (SERNAGEOMIN), 2011*; *Neira, 2012*).

In a mine site, the mineral of interest constitutes only a small fraction of the mined material (*Wills & Finch, 2015*), because of this the mining process generates large volumes of waste, originating a great amount of tailings and mine waste in general, which contain a high variety of heavy metals and diversity of concentration levels (*Burges, Epelde & Garbisu, 2015*; *Pourret et al., 2016*; *Lam et al., 2016*, *2017*). This renders many hectares of soil unsuitable for agriculture and generates highly contaminated soils, in which substances will move depending on the physicochemical properties of the substrate and on the climate conditions of the area in which the deposit is located (*Alloway, 2013*; *Chadwick, Highton & Lindman, 2013*; *Li et al., 2014*; *Pandey, Agrawal & Singh, 2016*; *Antoniadis et al., 2017*).

Closing a mine using low technology and without having an adequate plan that would enable to ensure the health and safety of the people and the environment brings about socio-environmental, financial and economic liabilities, affecting mainly communities close to where the mining sites are or have been, or where processes associated with extraction and processing of minerals are carried out, including electric generation, mineral transportation, and waste disposal, among others (*González, Stotz & Lancellotti, 2014*; *Marnika, Christodoulou & Xenidis, 2015*; *Ettler, Quantin & Kersten, 2016*; *Lechner, Kassulke & Unger, 2016*; *Schoenberger, 2016*; *Espinoza & Morris, 2017*; *Garcia et al., 2017*).

Abandoned and/or paralyzed mining sites that are distributed throughout the country constitute potential sources of air, water, and soil pollution; as well as potential harm to the population's environment and health (*Li et al., 2014*; *Diami, Kusin & Madzin, 2016*; *Gutiérrez, Mickus & Camacho, 2016*; *Pareja-Carrera, Mateo & Rodríguez-Estival, 2014*; *Carkovic et al., 2016*; *Obiora, Chukwu & Davies, 2016*; *Antoniadis et al., 2017*; *Ghorbani & Kuan, 2017*; *Christou et al., 2017*; *Espinoza & Morris, 2017*; *Unger, 2017*). It is imperative to face these issues, this requires identifying the potentially contaminated sites, the concentration, and variability of contaminants present in them, and also identifying the potential "victims" of these liabilities. In addition, it is necessary to consider the availability of technological and financial resources to address this new challenge

generated by a mining industry that did not have the vision of a sustainable development, developing overexploitation and damage of resources, applying poor management practices and inadequate technology (*Oyarzún & Oyarzún, 2011*; *Lam et al., 2016*; *Christou et al., 2017*; *Espinoza & Morris, 2017*; *Unger, 2017*).

The first regulations for estimating the degree of soil contamination were created in the Netherlands (*Boekhold, 2008*). This legislation provides procedures and standards for the short-term sanitation of contaminated soils. The law established limits depending on several factors: the nature and concentration of the contaminants and the conditions of the place where the contaminants are (e.g., soil characteristics).

In Chile, there have been several episodes of environmental impact on the marine environment due to the presence of mine tailings deposits which hamper port activities, generate geomorphological modifications on the coast and affect coastal ecosystems and recreational activities (*Castilla & Nealler, 1978*; *Castilla, 1983*; *Salamanca, Jara & Rodríguez, 2004*; *Ramírez et al., 2005*; *Besaury et al., 2013*; *Valladares et al., 2013*; *Dold, 2014*; *Contreras-Porcia et al., 2017*; *Monsalve et al., 2017*). It is necessary to give a solution to these liabilities as soon as possible, for they have generated chronic problems for the population, posing an even more serious threat to future generations.

Given the above, it would be very useful to have a tool that allowed evaluating whether a tailing requires intervention or not. The aim of this study is to develop a methodology, based on the Dutch regulations (*Dutch Ministry of Infrastructure and the Environment, 2013b*), that allows classifying the tailings according to their intervention requirements as: (1) it does not require intervention; (2) it requires intervention; (3) intervention is conditional on the availability of more information.

The methodology developed can be applied by means of a graphical method, which is used in case of having data on metal concentration and soil composition (in terms of its percentage of clay), or through a method based on conditional and unconditional threshold values (intervention thresholds) that only requires knowing the data of metal concentration in the soil. This allows applying the method even in situations where all the information required is not available. The methodology has been designed in such a way that future updates of the Dutch regulations are easily applicable.

A methodology as the one presented here will allow estimating if it is necessary to apply an intervention on the tailings found throughout Chile, as well as prioritizing those that require a more urgent intervention. Having a tool as the one presented in this work is vital for all those sites where there are tailings and the soil quality regulations are weak, or worse still, non-existent. It is important to note that the Dutch legislation, thanks to its rigorous foundation, is applied in Chile by the National Service of Geology and Mining (SERNAGEOMIN), as well as in other countries (*Milenkovic, Damjanovic & Ristic, 2005*; *Swartjes et al., 2012*).

## MATERIALS AND METHODS

### Methodological proposal

The proposal is based on the Dutch legislation for the regulation of soil quality (*Dutch Ministry of Infrastructure and the Environment, 2013b*). In particular, this law provides

intervention values for different metals. The intervention values are threshold concentrations above which it is considered that the soil presents a serious case of contamination. Above the intervention values, the functionality of the soil for human, animal, or plant life is seriously affected or complicated. In particular, the 2013 revised version of the Dutch standard (*Dutch Ministry of Infrastructure and the Environment, 2013a*) will be used for the base values.

The selection of this regulation was based on the following aspects: (1) Dutch legislation provides a mathematical formula that allows adapting its use depending on the nature of the soil; (2) it is one of the most stringent regulations for the evaluation of soils (*Macklin et al., 2003*); and (3) the Dutch standard has been used for almost four decades, which makes it one of the longest-running standards in this field.

Although the standard presents some limitations, it has been widely used in the literature since its creation and it allows evaluating and filtering out the sites that do not require intervention. Some recent examples of the application can be found in the study of metal concentration in agricultural soils (*Kelepertzis, 2014*), urban soils (*Darko et al., 2017*), and mine soils (*Bempah & Ewusi, 2016*). It should be noted that these applications have been carried out in different countries with soils of diverse characteristics.

In general, to evaluate the soil quality according to the Dutch guidelines, the standard intervention values must be converted to values that correspond to the characteristics of the soil to be evaluated. The intervention values are then compared with the concentration found in the soil. The characterization of the soil is done by measuring the percentage of clay and the organic matter present in the soil. The soil intervention value (SIV) is calculated through the formula shown in Eq. (1) (*Dutch Ministry of Infrastructure and the Environment, 2013a*).

$$\text{SIV} = \text{SSIV} \cdot \frac{A + B \cdot x_A + C \cdot x_M}{A + 25 \cdot B + 10 \cdot C} \tag{1}$$

Where each term of the equation is defined as follows:

- SSIV corresponds to the Standard Soils Intervention Value. SSIV is a value defined for a soil with 10% organic matter and 25% clay for each element. Table 1 presents the values for each element.
- Constants $A$, $B$, and $C$ correspond to parameters based on the characteristics of each element. Table 1 presents the values of these constants for some relevant metals.
- The variable $x_A$ corresponds to the percentage of clay in the substrate that is being evaluated, expressed as a number between 0 and 100. If the clay content is less than 2% then $x_A$ is assigned the value 2 (that is, the lowest value it can take is 2%).
- The variable $x_M$ corresponds to the percentage of organic matter in the substrate that is being evaluated, expressed as a number between 0 and 100. If the content of organic matter is less than 2% then $x_M$ is assigned the value 2 (that is, the lowest value it can take is 2%).

**Table 1 Parameters of the equation for the calculation of SSIV for each element (source: *Dutch Ministry of Infrastructure and the Environment, 2013a*).**

| Element | A | B | C | SSIV (mg/kg) |
|---|---|---|---|---|
| Arsenic | 15 | 0.4 | 0.4 | 76 |
| Cadmium | 0.4 | 0.007 | 0.021 | 13 |
| Mercury | 0.2 | 0.0034 | 0.0017 | 4 |
| Lead | 50 | 1 | 1 | 530 |
| Nickel | 10 | 1 | 0 | 100 |
| Zinc | 50 | 3 | 1.5 | 720 |
| Copper | 15 | 0.6 | 0.6 | 190 |
| Chromium | 50 | 2 | 0 | 78 |

**Note:**
Each row shows the value of the corresponding parameter for each element.

In the methodology proposed, two assumptions are made:

1. It is assumed that the data have been previously gathered, that the mineral concentration in the soil is available and, optionally, the percentage of clay in the soil.

2. It is assumed that the tailings do not have organic matter, or its percentage is equal to or less than 2%.

In case that both the concentration of metal in the soil and the percentage of clay are available, a graphical method can be directly applied to evaluate the necessity of intervention in a soil. On the contrary, if the clay percentage is not available, the methodology proposed allows using conditional and unconditional intervention thresholds defined in this work to determine the intervention requirements and prioritize the sites.

On the other hand, it must be mentioned that the proposal in this work can be generalized, since, although the methodology proposed has been developed for soils containing mine tailings, a similar strategy could be applied for soils of similar characteristics or that accept assumptions of similar nature.

## Intervention values and graphical method

The lowest value that each percentage $x_A$ and $x_M$ can be assigned is 2. In particular, the composition of tailing deposits guarantees that the percentage of organic matter is negligible (i.e., close to 0); thus, according to the conditions of the method, it is assumed that for all the soils considered in this work $x_M = 2$.

Bearing in mind all previous observations, in this work a referential table of the intervention values has been built. These are presented in Table 2. SIVs have been calculated using the method provided by the Dutch guidelines under the supposition that the organic matter percentage in a tailing is negligible ($\leq 2\%$). If the concentration in mg/kg exceeds the values indicated in this table for the composition of a given soil, then the tailings deposit must be intervened.

In case that the clay percentage in the soil is not found in Table 2, a linear interpolation can be used to obtain the result. This will produce the correct result (because the base

**Table 2 Referential table of intervention values (SIV) of each element for different soils according to clay percentage assuming organic matter is ≤2% (source: own elaboration).**

| SIV (mg/kg) | | Element | | | | | | | |
|---|---|---|---|---|---|---|---|---|---|
| | | As | Cd | Hg | Pb | Ni | Zn | Cu | Cr |
| Percentage of clay | 2 | 43.50 | 7.55 | 2.78 | 336.71 | 34.29 | 303.43 | 91.83 | 42.12 |
| | 5 | 46.65 | 7.90 | 2.92 | 355.41 | 42.86 | 349.71 | 101.33 | 46.80 |
| | 10 | 51.89 | 8.48 | 3.14 | 386.59 | 57.14 | 426.86 | 117.17 | 54.60 |
| | 15 | 57.13 | 9.06 | 3.37 | 417.76 | 71.43 | 504.00 | 133.00 | 62.40 |
| | 20 | 62.37 | 9.64 | 3.59 | 448.94 | 85.71 | 581.14 | 148.83 | 70.20 |
| | 25 | 67.61 | 10.22 | 3.82 | 480.12 | 100.00 | 658.29 | 164.67 | 78.00 |
| | 30 | 72.86 | 10.80 | 4.05 | 511.29 | 114.29 | 735.43 | 180.50 | 85.80 |
| | 35 | 78.10 | 11.38 | 4.27 | 542.47 | 128.57 | 812.57 | 196.33 | 93.60 |
| | 40 | 83.34 | 11.96 | 4.50 | 573.65 | 142.86 | 889.71 | 212.17 | 101.40 |
| | 45 | 88.58 | 12.54 | 4.72 | 604.82 | 157.14 | 966.86 | 228.00 | 109.20 |
| | 50 | 93.82 | 13.12 | 4.95 | 636.00 | 171.43 | 1044.00 | 243.83 | 117.00 |
| | 55 | 99.06 | 13.70 | 5.17 | 667.18 | 185.71 | 1121.14 | 259.67 | 124.80 |
| | 60 | 104.30 | 14.28 | 5.40 | 698.35 | 200.00 | 1198.29 | 275.50 | 132.60 |
| | 65 | 109.54 | 14.85 | 5.62 | 729.53 | 214.29 | 1275.43 | 291.33 | 140.40 |
| | 70 | 114.79 | 15.43 | 5.85 | 760.71 | 228.57 | 1352.57 | 307.17 | 148.20 |
| | 75 | 120.03 | 16.01 | 6.07 | 791.88 | 242.86 | 1429.71 | 323.00 | 156.00 |
| | 80 | 125.27 | 16.59 | 6.30 | 823.06 | 257.14 | 1506.86 | 338.83 | 163.80 |
| | 85 | 130.51 | 17.17 | 6.52 | 854.24 | 271.43 | 1584.00 | 354.67 | 171.60 |
| | 90 | 135.75 | 17.75 | 6.75 | 885.41 | 285.71 | 1661.14 | 370.50 | 179.40 |
| | 95 | 140.99 | 18.33 | 6.97 | 916.59 | 300.00 | 1738.29 | 386.33 | 187.20 |
| | 100 | 146.23 | 18.91 | 7.19 | 947.77 | 314.29 | 1815.44 | 402.16 | 195 |

**Note:**
Each row of this table presents the intervention values (SIV) of each element, depending on the percentage of clay in the soil and assuming a negligible amount of organic matter.

calculation model is linear). If the clay percentage is less than 2%, it must be assumed that it takes the value of 2% (in accordance with the Dutch guidelines), so it must not be extrapolated.

The results of Table 2 are graphically represented for arsenic in Fig. 1 as an example. The clay percentage is on the abscissa axis and the concentration of the corresponding element in mg/kg is on the ordinate axis. These graphs show the straight line determined by the formula of intervention value for tailings (intervention threshold). Two zones can be observed in the graph: the zone above the intervention threshold that indicates the necessity of intervention and the zone below that represents the safe zone that does not require immediate intervention.

It should be noted that the intervention threshold is given by a line of positive slope; this suggests that for a given a value of the element concentration, then all the clay percentages above a certain threshold will not require an intervention (i.e., they will be in the safe zone). The threshold for this can be found graphically by tracing a horizontal line at the given concentration and finding the point where it intersects with the

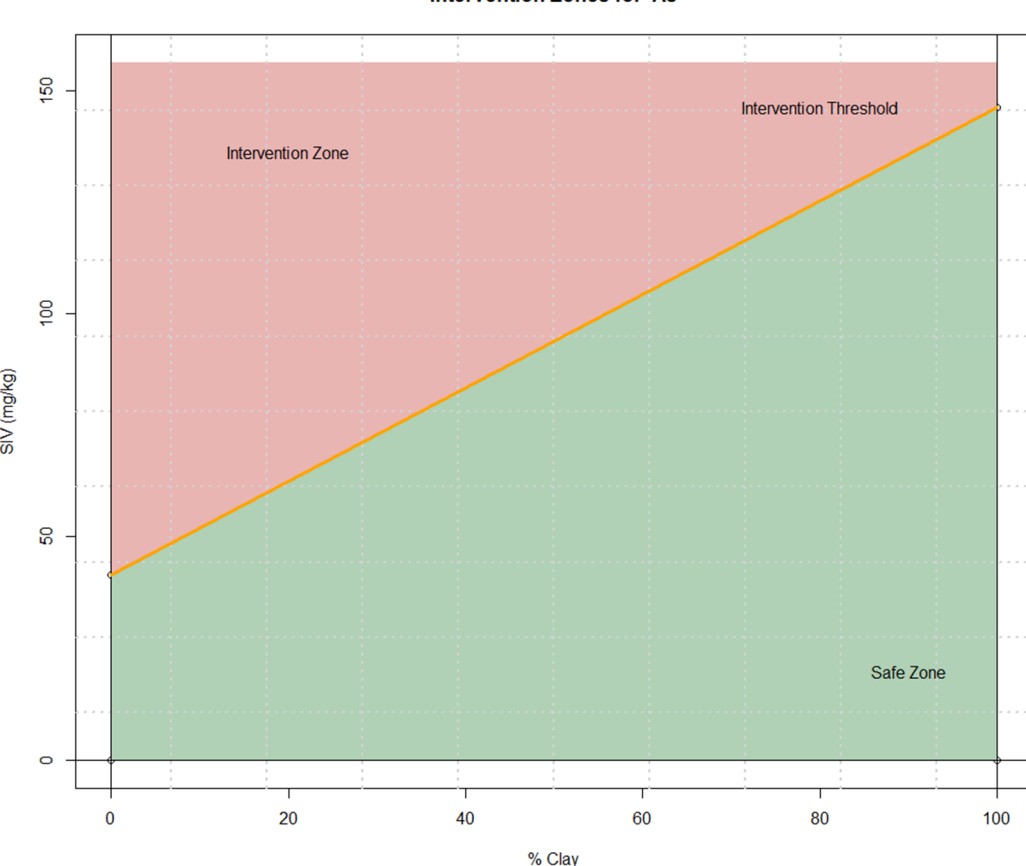

**Figure 1 Graphs of intervention regions for arsenic source: own elaboration.** Each graph represents the intervention zones according to the parameters of each element. The separating line corresponds with the intervention threshold.

intervention threshold inside the 0–100 range of the clay percentage. However, it is important to note that for some values of the concentration there will be no intersection; in fact, if the horizontal line lies above the intervention threshold line, then it is always necessary to intervene. On the other hand, if the horizontal line is below the intervention threshold line it will always be in the safe zone. These observations provide the motivation for the definition of the threshold factor and adjusted threshold factor in the next sections, which can be seen as a simpler quantitative alternative to the graphical methods.

## Threshold factor

This work defines the concept of threshold factor $C_F$, corresponding to the minimum percentage of clay acceptable in function of the concentration of the element measured (according to the parameters of the Dutch standard). If the real percentage of clay in the soil exceeds this value, then the intervention of the soil will not be required. In case the clay percentage is lower than the threshold factor then the soil will require intervention.

**Table 3 Summary of intervention cases with respect to the threshold factor.**

| Case | Condition | Description | Required actions | Subcases | Additional conditions |
|---|---|---|---|---|---|
| No intervention | $C_F \leq 0$ | The tailing deposit does not require intervention, regardless of the soil composition. | None | None | None |
| Conditional intervention | $0 < C_F < 100$ | The tailing deposit may or may not require intervention; this depends on the soil composition. | Determine the clay percentage $x_A$. | Intervention not required, it is not necessary to intervene the soil because it is under the intervention value specified for this type of soil. | $x_A > C_F$ |
| | | | | Intervention required, the soil must be intervened because it exceeds or equals the intervention value specified for this type of soil. | $x_A \leq C_F$ |
| Unconditional intervention | $C_F \geq 100$ | The tailing deposit requires intervention, regardless of the soil composition. | Prepare an intervention plan for the site. | None | None |

**Note:**
Each row describes a different case depending on the value of the threshold factor.

The threshold factor is obtained by setting $x_M = 2$ (because it is assumed that the organic matter content is negligible) and solving the SIV equation for $x_A$. From this procedure, the following equation is obtained:

$$C_F = \frac{\text{SIV}}{\text{SSIV}} \cdot \left( \frac{A + 25 \cdot B + 10 \cdot C}{B} \right) - \frac{(A + 2C)}{B} \qquad (2)$$

Note that although this formula can deliver values lower than 0 or higher than 100, these have no sense physically. In fact, these values are utilized as limits to determine if the tailings deposit does not require intervention or if the intervention is strictly necessary, regardless of the real percentage of clay in the soil. The threshold factor $C_F$ facilitates the analysis of the tailings deposits by the considerations shown in Table 3.

Thus, it is recommended that samples are obtained to evaluate the clay percentage of the soils in the tailings deposits that have a threshold factor between 0 and 100 (conditional intervention).

## Adjusted threshold factor

It should be noted that the threshold factor in its original definition brings about problems of scale when converting the results obtained with real values into a graph. In order to simplify the graphical analysis of the results, the adjusted threshold factor ($AC_F$) is proposed in Eq. (3). This minimizes the problems of scale and facilitates interpretation.

$$AC_F = \text{sign}(C_F) \cdot \log(1 + \text{abs}(C_F)) \qquad (3)$$

**Table 4 Summary of intervention cases with respect to the adjusted threshold factor.**

| Case | Condition | Priority | Required actions |
|---|---|---|---|
| No intervention | $AC_F \leq 0$ | None | None |
| Unlikely conditional intervention | $0 < AC_F \leq 1$ | Low | If possible, determine the clay percentage $x_A$ to find if intervention is required. |
| Conditional intervention | $1 < AC_F < 2$ | Medium | Determine the clay percentage $x_A$ and find if intervention is required. |
| Unconditional intervention | $AC_F \geq 2$ | High | Prepare an intervention plan for the site. |

**Note:**
Each row describes a different case depending on the value of the adjusted threshold factor.

It should be noted that this is similar to a logarithmic scale, but it admits negative values. The evaluation by means of $AC_F$ is carried out as follows:

- If $AC_F \leq 0$ then it is not necessary to intervene (it corresponds to the cases where $C_F \leq 0$).
- If $AC_F \geq 2$ then it is necessary to intervene (it corresponds to the cases where $C_F \geq 100$). It must be noted that $AC_F = 2.004$ when $C_F = 100$, but for practical purposes the difference is negligible, and the analysis is much simpler in this way. These sites should have the highest priority of intervention.
- If $0 < AC_F \leq 1$ it corresponds to the cases in which the unadjusted threshold factor is between 0 and 10 approximately. In this case the need of intervention is unlikely, however it is, still considered as a conditional intervention. These sites should not be prioritized above the next ones.
- If $1 < AC_F < 2$ it corresponds to the cases in which the unadjusted threshold factor is between 10 and 100 approximately. In this case, the need of intervention is already more likely, and it is considered as a conditional intervention. These sites should have the next highest priority after unconditional interventions.

These cases are summarized in Table 4 which can be seen as the adjusted version of Table 3.

Based on a similar reasoning to the unlikely conditional intervention case, high values of the adjusted conditional factor (i.e., close to 2) could probably be taken as sites with a high probability of requiring an intervention, thus it would be recommendable to act as if for every site with an $AC_F \geq 2-\varepsilon$ for some small value $\varepsilon > 0$ was actually an unconditional intervention. Note that this last recommendation is more of a heuristic to reduce the extra resources that would be needed to take another sample to determine the real clay percentage. This is especially important if there are more sites in a conditional intervention state than available resources for sampling. In light of this, the value of $\varepsilon$ should be chosen carefully.

## Andacollo mine tailings in Chile

According to the survey carried out in December 2016 by the SERNAGEOMIN, in Chile there are 696 tailings deposits, catalogued as active (16.1%), inactive (62.6%), and abandoned (21.3%), distributed from the Tarapacá region up to the Metropolitan region. (Tarapacá 1.00%, Antofagasta 6.18%, Atacama 22.27%, Coquimbo 52.87%, Valparaíso

10.49%, Bernardo O'Higgins 2.59%, Maule 0.43%, Aysén 0.72%, and Metropolitan region 3.45%).

Of particular interest is the Coquimbo region for its great number of tailing deposits compared to the other regions of the country. The Coquimbo region has been an almost continuous exploited source of Cu, Au, and Hg in Chile for centuries. In spite of this, the communities living in this zone are still underdeveloped and suffer from the extended contamination produced by the inefficient treatment of mine tailings and wastes (*Higueras et al., 2004*). It must be noted that bioremediation plans exist for the commune; however, there are a number of problems, such as a lack of a regulatory legal framework and operational issues, which prevent their implementation (*Leiva & Morales, 2013*).

This case study focuses on the commune of Andacollo, in particular, the utilized data corresponds to the geochemical characterization of tailings deposits carried out by SERNAGEOMIN in the commune of Andacollo in the Coquimbo Region, Chile (*National Geology and Mining Service of Chile (SERNAGEOMIN), 2017*). In particular, 81 samples have been analyzed corresponding to 22 tailings deposits. There have been previous studies to assess the contamination and risks in the commune of Andacollo, such as *Higueras et al. (2004)* where a general environmental analysis was carried out, detecting significant contamination of the surrounding landscape due to decades of inefficient treatment of waste-rock stockpiles and flotation tailings.

The present work focuses on the elements considered critical for the environment, presenting the analysis carried out to the following elements of environmental relevance related to mining activity: Hg, Cd, Pb, As, Cu, Ni, Zn, Cr. Andacollo is located in the Coquimbo region of Chile and is about 57 km to the southeast of La Serena. It is situated at latitude 30°12′00″ south and longitude 71°05′00″ east. It covers an area of about 310 km$^2$ and is bounded on the south by Ovalle, northeast by the Commune of Vicuña, and southeast by the Commune of Río Hurtado and, west by the Commune of Coquimbo, and north by the Commune of La Serena. It is also home to several mining activities (*Higueras et al., 2004*).

To use the graphical method described in the section 'Intervention values and graphical method', it is necessary to have data about the percentage of clay and the concentration of the element of interest. Using this information, the sample must be located in the corresponding graph to see in which zone it lies. However, since this method requires the value of clay percentage, it is not possible to apply it directly to the data provided by SERNAGEOMIN. Thus, the threshold values approach is used for this data set. The results obtained for the different elements studied are presented and discussed. Specifically, the state of each tailing is analyzed based on the criterion defined by the threshold factor.

The results obtained for the different elements studied are presented and discussed. The samples have been identified following the identification number of the tailing deposit and its origin that can be from the tailings pond (TP), the sediments (S) or the wall (W). The concentration values reported as lower than a certain number $n$ (in mg/kg) were assigned the value $n$ mg/kg to obtain an estimate corresponding to the worst case. Subsequently, the same scheme is applied if the data have that format. Specifically, the state of each tailing is analyzed based on the criterion defined by the threshold factor.

**Table 5  Summary of results for all the studied metals.**

| Element | Deposit status | $AC_F \leq 0$ (no intervention) | $0 < AC_F < 2$ (conditional intervention) | $AC_F \geq 2$ (unconditional intervention) | Total |
|---|---|---|---|---|---|
| As | Active | 10 | 1 | 0 | 11 |
| | Inactive | 18 | 0 | 4 | 22 |
| | Abandoned | 48 | 0 | 0 | 48 |
| Cd | Active | 9 | 2 | 0 | 11 |
| | Inactive | 10 | 12 | 0 | 22 |
| | Abandoned | 30 | 18 | 0 | 48 |
| Pb | Active | 11 | 0 | 0 | 11 |
| | Inactive | 18 | 4 | 0 | 22 |
| | Abandoned | 48 | 0 | 0 | 48 |
| Ni | Active | 0 | 11 | 0 | 11 |
| | Inactive | 0 | 22 | 0 | 22 |
| | Abandoned | 0 | 48 | 0 | 48 |
| Hg | Active | 10 | 1 | 0 | 11 |
| | Inactive | 20 | 2 | 0 | 22 |
| | Abandoned | 39 | 8 | 1 | 48 |
| Cu | Active | 0 | 2 | 9 | 11 |
| | Inactive | 0 | 6 | 16 | 22 |
| | Abandoned | 1 | 8 | 39 | 48 |
| Zn | Active | 10 | 1 | 0 | 11 |
| | Inactive | 18 | 4 | 0 | 22 |
| | Abandoned | 48 | 0 | 0 | 48 |
| Cr | Active | 1 | 10 | 0 | 11 |
| | Inactive | 6 | 10 | 6 | 22 |
| | Abandoned | 13 | 35 | 0 | 48 |
| Total | | | | | 648 |

Note:
This table presents the information regarding to the necessity of intervention results for the tailings according to the adjusted threshold factor for the different metals grouped by tailing deposit status (active, inactive, or abandoned).

# RESULTS

## Arsenic

The results obtained for each sample available in the SERNAGEOMIN data are summarized in Table 5. It is noted that for most samples, arsenic levels are low enough to guarantee that intervention is not necessary at the moment. On the other hand, there are four samples that suggest an unconditional intervention, regardless of the percentage of clay in the soil, while there is only one sample that indicates a conditional need of intervention in a tailings deposit. Only in the latter case, it would be necessary to obtain the real value of the clay percentage of the soil. It must be noted that the elements that are shown in pairs with similar values come from the same deposit, but from different samples, due to which they exhibit a similar behavior. This same pattern is repeated in the subsequent analyses.

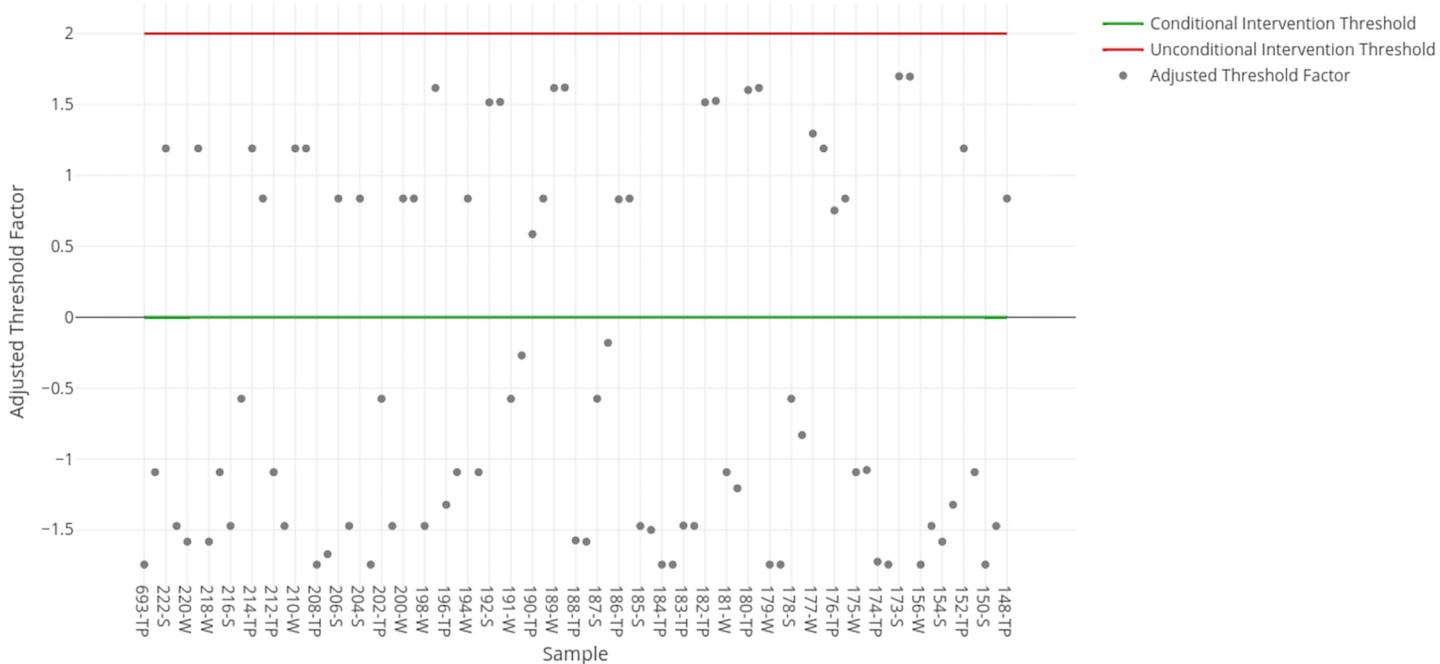

**Figure 2 Adjusted threshold factor for cadmium.** Each data point indicates the value of the adjusted threshold factor of a tailing sample for cadmium. The red line indicates the unconditional intervention threshold and the green line indicates the conditional intervention threshold. The data points between these lines are considered for a conditional intervention, while the points above the red line are considered for an unconditional intervention.

In particular, the samples 191-W, 191-W-2, 190-TP, and 190-TP-2 require unconditional intervention. These samples are associated with the ARIZONA 1 and ARIZONA 2 tailing deposits. Sample 152-TP suggests that the deposit SANTA TERESITA 2 requires conditional intervention.

## Cadmium

The threshold factors obtained for each sample are shown in Fig. 2. The results obtained for each sample available in the SERNAGEOMIN data are summarized in Table 5. In this case, it should be noted that none of the samples suggests an unconditional intervention. Nevertheless, a considerable number of samples suggests a conditional intervention, due to which it is important to determine the corresponding percentage of clay to evaluate the course of actions needed in those deposits.

## Lead

The results obtained for each sample available in the SERNAGEOMIN data are summarized in Table 5. In this case, there are four samples that indicate that a conditional intervention is required, hence it is necessary to obtain the corresponding percentage of clay. Regarding the other cases, it can be seen that the analysis of most samples indicates that the deposits do not require any intervention.

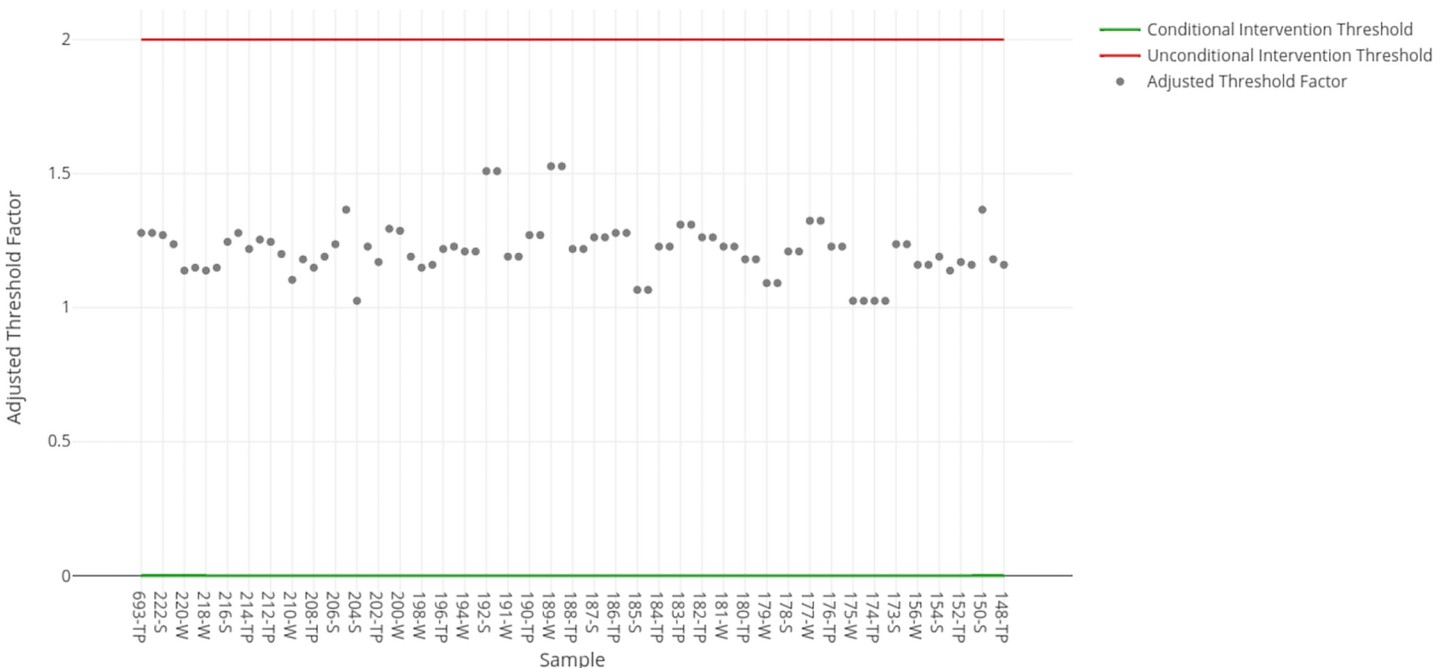

**Figure 3 Adjusted threshold factor for nickel.** Each data point indicates the value of the adjusted threshold factor of a tailing sample for nickel. The red line indicates the unconditional intervention threshold and the green line indicates the conditional intervention threshold. The data points between these lines are considered for a conditional intervention, while the points above the red line are considered for an unconditional intervention.                                                                   

In particular, the samples 191-W, 191-W-2, 190-TP, and 190-TP-2 require conditional intervention. These samples are associated with the ARIZONA 1 and ARIZONA 2 tailing deposits, the same deposits that required unconditional intervention for arsenic.

## Nickel

The threshold factors obtained for each sample are shown in Fig. 3. The results obtained for each sample available in the SERNAGEOMIN data are summarized in Table 5. The particular case of Nickel is different from the previous ones since none of the extreme values observed above are present here. For this criterion, all the tailings are classified as requiring conditional intervention, due to which it is necessary to determine the percentage of clay to decide whether there should be intervention or not.

## Mercury

The threshold factors obtained for each sample are shown in Fig. 4. The results obtained for each sample available in the SERNAGEOMIN data are summarized in Table 5. Regarding the results obtained for mercury, it is necessary to note that a great majority are below intervention values, due to which in the short term it is not necessary to carry out interventions on them. Nevertheless, there is a sample that indicates the need for
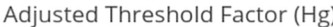

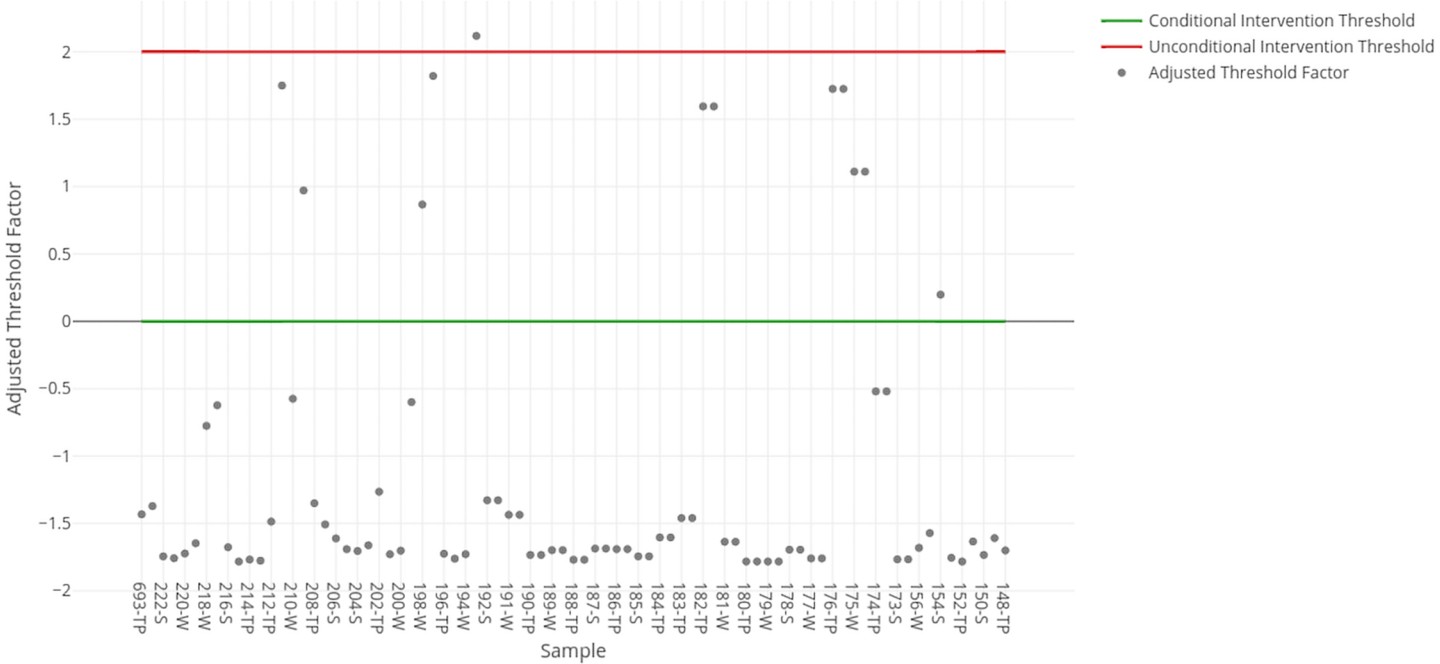

**Figure 4 Adjusted threshold factor for mercury.** Each data point indicates the value of the adjusted threshold factor of a tailing sample for mercury. The red line indicates the unconditional intervention threshold and the green line indicates the conditional intervention threshold. The data points between these lines are considered for a conditional intervention, while the points above the red line are considered for an unconditional intervention.

unconditional intervention (corresponding to an abandoned deposit) and some that indicate conditional intervention with the available data.

## Copper

The threshold factors obtained for each sample are shown in Fig. 5. The results obtained for each sample available in the SERNAGEOMIN data are summarized in Table 5. The analysis of the data shows the necessity of intervention of the tailings deposits regarding copper concentration. There is only one sample that indicates that intervention is not needed, and it corresponds to an abandoned deposit, all the other cases require intervention to some degree, be it conditional or unconditional.

## Zinc

The results obtained for each sample available in the SERNAGEOMIN data are summarized in Table 5. In this case, there are no samples that suggest an unconditional intervention. There are five samples that indicate that conditional intervention is required, which correspond mainly to inactive deposits. The other samples correspond in its majority to deposits that do not require intervention. Regarding the five that require conditional intervention, it is necessary to obtain the real values of the percentage of clay to define whether intervention is needed.

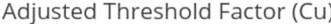

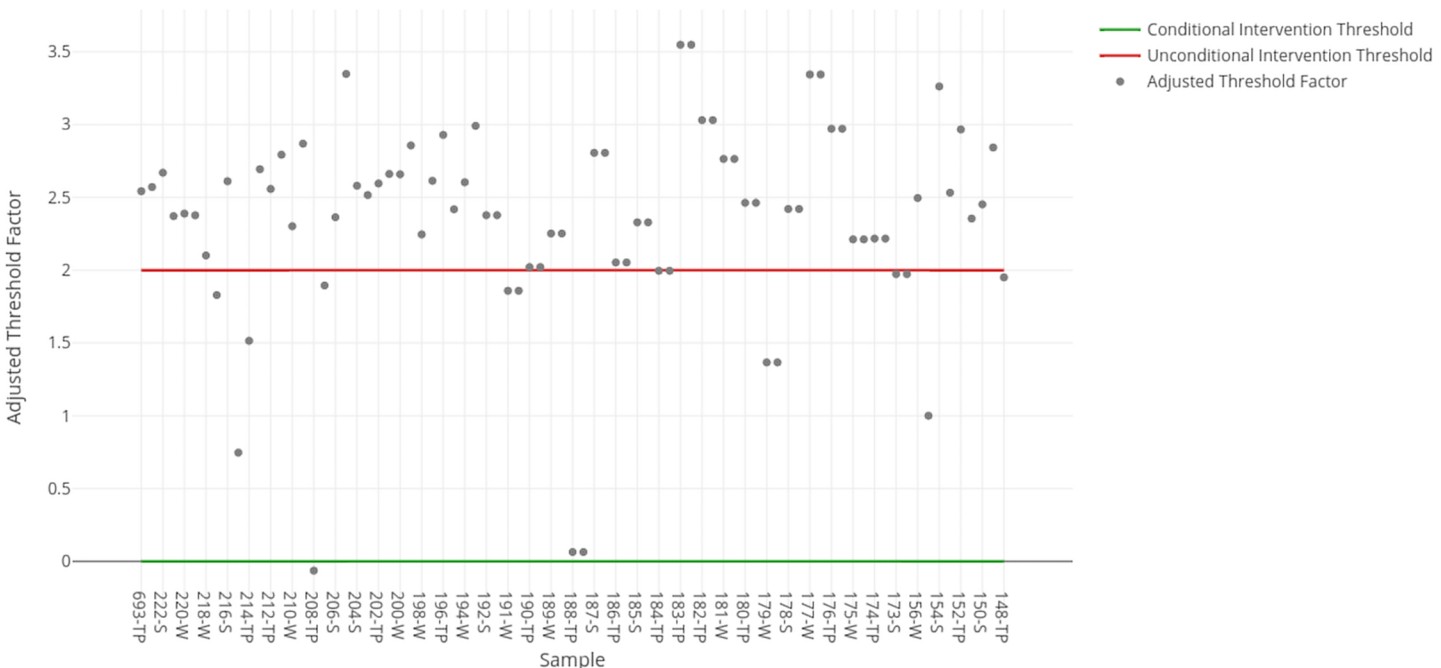

**Figure 5 Adjusted threshold factor for copper.** Each data point indicates the value of the adjusted threshold factor of a tailing sample for copper. The red line indicates the unconditional intervention threshold and the green line indicates the conditional intervention threshold. The data points between these lines are considered for a conditional intervention, while the points above the red line are considered for an unconditional intervention.

In particular, the samples 191-W, 191-W-2, 190-TP, 190-TP-2, and 154-S require conditional intervention. These samples are associated with the ARIZONA 1, ARIZONA 2, and SANTA TERESITA 2 tailing deposits.

### Chromium

The threshold factors obtained for each sample are shown in Fig. 6. The results obtained for each sample available in the SERNAGEOMIN data are summarized in Table 5. For chromium, it can be observed that there are six samples of inactive deposits that indicate the necessity of unconditional intervention. Most samples suggest only a conditional intervention, while the rest would not require intervention in the short term.

## DISCUSSION

### General evaluation

Out of the 696 deposits surveyed by *National Geology and Mining Service of Chile (SERNAGEOMIN) (2016)*, 368 are found in the Coquimbo region, distributed in 13 communes, one of them being the commune of Andacollo, which concentrates the highest number of tailing deposits in the region. According to the survey by *National Geology and Mining Service of Chile (SERNAGEOMIN) (2016)*, out of the 368 deposits in the

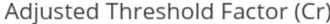

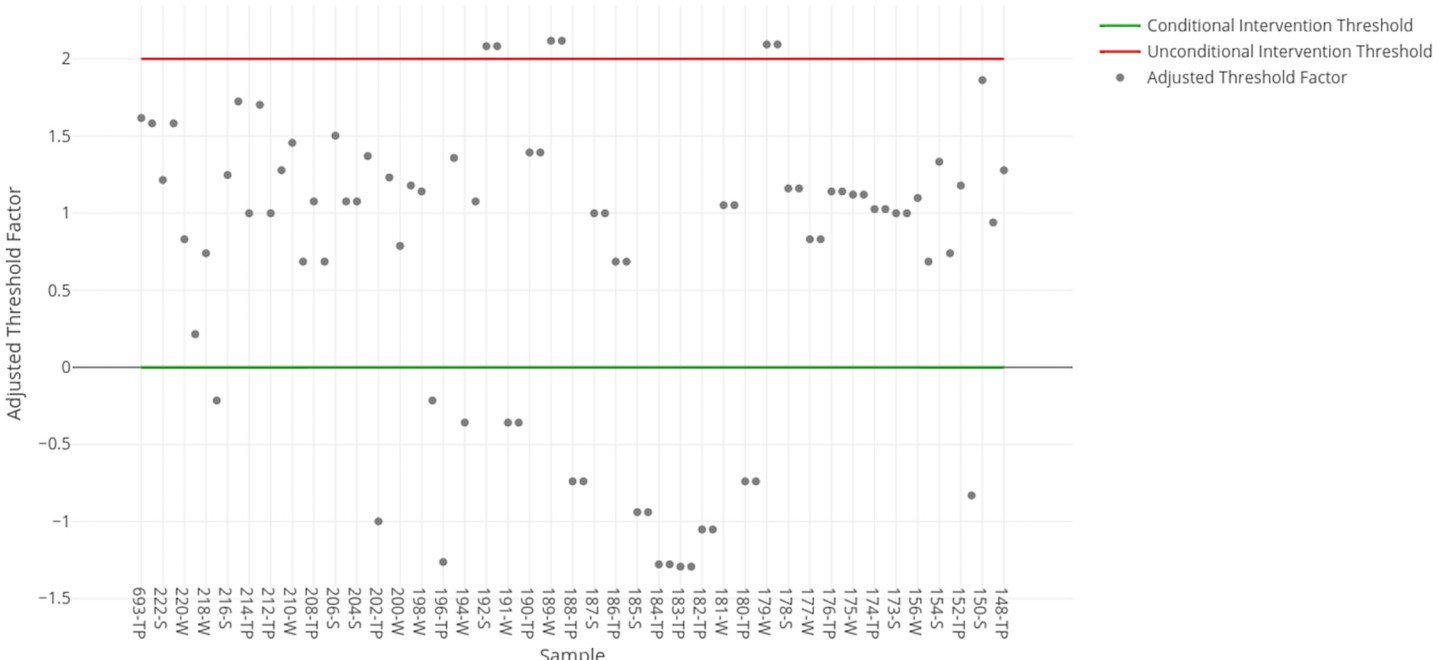

**Figure 6 Adjusted threshold factor for chromium.** Each data point indicates the value of the adjusted threshold factor of a tailing sample for chromium. The red line indicates the unconditional intervention threshold and the green line indicates the conditional intervention threshold. The data points between these lines are considered for a conditional intervention, while the points above the red line are considered for an unconditional intervention.

region, 118 are found in the commune of Andacollo, which are then classified according to their status as abandoned (35), active (eight), and inactive (75).

Nine tailing deposits are found in urban areas in the commune of Andacollo, which were abandoned in the 1950's. That is to say, they have coexisted with the population for 70 years, constituting a potential risk for them and the environment. Given the high costs associated with the required interventions, it is necessary to rank the tailings to intervene according to priority. The information regarding the current tonnage of the tailings is scarce; however, the information available corresponds to the tonnages of tailings authorized for each deposit, which fluctuate between 500,000 and 416,000,000 tons.

The deposits that constitute major problems correspond to those that are abandoned, since they do not have a responsible organization or person to take charge of them, being the State of Chile the one that must have the necessary resources to face this great challenge. The authorized values for the abandoned tailings in the commune of Andacollo are between 500 and 227,850 tons.

Having carried out the corresponding analysis for each element, the summary of results obtained for each tailings deposit status (Active, Inactive, and Abandoned) is shown. Table 6 presents a summary of the results obtained for each sample of the SERNAGEOMIN data for active and inactive deposits. It should be noted that in almost all cases of active deposits it is necessary to carry out an unconditional intervention in each

**Table 6 Summary of results by sample for ACTIVE deposits.**

| Deposit status | ID | Cr | Zn | Ni | Pb | Hg | Cu | Cd | As |
|---|---|---|---|---|---|---|---|---|---|
| ACTIVE | 148-TP | CND | NO | CND | NO | NO | CND | CND | NO |
| | 149-W | CND | NO | CND | NO | NO | **YES** | NO | NO |
| | 150-S | CND | NO | CND | NO | NO | **YES** | NO | NO |
| | 151-W | NO | NO | CND | NO | NO | **YES** | NO | NO |
| | 152-TP | CND | NO | CND | NO | NO | **YES** | CND | CND |
| | 153-W | CND | NO | CND | NO | NO | **YES** | NO | NO |
| | 154-S | CND | CND | CND | NO | CND | **YES** | NO | NO |
| | 155-TP | CND | NO | CND | NO | NO | CND | NO | NO |
| | 156-W | CND | NO | CND | NO | NO | **YES** | NO | NO |
| | 692-TP | CND | NO | CND | NO | NO | **YES** | NO | NO |
| | 693-TP | CND | NO | CND | NO | NO | **YES** | NO | NO |
| INACTIVE | 176-TP | CND | NO | CND | NO | CND | **YES** | CND | NO |
| | 176-TP-2 | CND | NO | CND | NO | CND | **YES** | CND | NO |
| | 177-W | CND | NO | CND | NO | NO | **YES** | CND | NO |
| | 177-W-2 | CND | NO | CND | NO | NO | **YES** | CND | NO |
| | 178-S | CND | NO | CND | NO | NO | **YES** | NO | NO |
| | 178-S-2 | CND | NO | CND | NO | NO | **YES** | NO | NO |
| | 179-W | **YES** | NO | CND | NO | NO | CND | NO | NO |
| | 179-W-2 | **YES** | NO | CND | NO | NO | CND | NO | NO |
| | 180-TP | NO | NO | CND | NO | NO | **YES** | CND | NO |
| | 180-TP-2 | NO | NO | CND | NO | NO | **YES** | CND | NO |
| | 181-W | CND | NO | CND | NO | NO | **YES** | NO | NO |
| | 181-W-2 | CND | NO | CND | NO | NO | **YES** | NO | NO |
| | 188-TP | NO | NO | CND | NO | NO | CND | NO | NO |
| | 188-TP-2 | NO | NO | CND | NO | NO | CND | NO | NO |
| | 189-W | **YES** | NO | CND | NO | NO | **YES** | CND | NO |
| | 189-W-2 | **YES** | NO | CND | NO | NO | **YES** | CND | NO |
| | 190-TP | CND | CND | CND | CND | NO | **YES** | CND | **YES** |
| | 190-TP-2 | CND | CND | CND | CND | NO | **YES** | CND | **YES** |
| | 191-W | NO | CND | CND | CND | NO | CND | NO | **YES** |
| | 191-W-2 | NO | CND | CND | CND | NO | CND | NO | **YES** |
| | 192-S | **YES** | NO | CND | NO | NO | **YES** | CND | NO |
| | 192-S-2 | **YES** | NO | CND | NO | NO | **YES** | CND | NO |

Notes:
"NO": no intervention, "YES": unconditional intervention, and "CND": conditional intervention.
Each row corresponds to a tailing sample from active deposits, the columns show if this sampling suggests a conditional intervention, an unconditional intervention, or no intervention according to the adjusted threshold factor.

deposit due to copper concentrations. If the particular case of copper is not considered, conditional intervention is required in all tailings. It should be noted that in almost all cases an unconditional intervention in the deposit is needed due to copper concentrations, although in contrast to the previous case, there are also cases that show a potentially problematic concentration of chromium or arsenic. At any rate, if the particular case of copper is not considered, conditional intervention is required in all tailings.

**Table 7 Summary of results by sample for Abandoned deposits.**

| ID | Cr | Zn | Ni | Pb | Hg | Cu | Cd | As |
|---|---|---|---|---|---|---|---|---|
| 173-S | CND | NO | CND | NO | NO | CND | CND | NO |
| 173-S-2 | CND | NO | CND | NO | NO | CND | CND | NO |
| 174-TP | CND | NO | CND | NO | NO | **YES** | NO | NO |
| 174-TP-2 | CND | NO | CND | NO | NO | **YES** | NO | NO |
| 175-W | CND | NO | CND | NO | CND | **YES** | NO | NO |
| 175-W-2 | CND | NO | CND | NO | CND | **YES** | NO | NO |
| 182-TP | NO | NO | CND | NO | CND | **YES** | CND | NO |
| 182-TP-2 | NO | NO | CND | NO | CND | **YES** | CND | NO |
| 183-TP | NO | NO | CND | NO | NO | **YES** | NO | NO |
| 183-TP-2 | NO | NO | CND | NO | NO | **YES** | NO | NO |
| 184-TP | NO | NO | CND | NO | NO | CND | NO | NO |
| 184-TP-2 | NO | NO | CND | NO | NO | CND | NO | NO |
| 185-S | NO | NO | CND | NO | NO | **YES** | NO | NO |
| 185-S-2 | NO | NO | CND | NO | NO | **YES** | NO | NO |
| 186-TP | CND | NO | CND | NO | NO | **YES** | CND | NO |
| 186-TP-2 | CND | NO | CND | NO | NO | **YES** | CND | NO |
| 187-S | CND | NO | CND | NO | NO | **YES** | NO | NO |
| 187-S-2 | CND | NO | CND | NO | NO | **YES** | NO | NO |
| 193-TP | CND | NO | CND | NO | SÍ | **YES** | NO | NO |
| 194-W | NO | NO | CND | NO | NO | **YES** | CND | NO |
| 195-S | CND | NO | CND | NO | NO | **YES** | NO | NO |
| 196-TP | NO | NO | CND | NO | NO | **YES** | NO | NO |
| 197-W | NO | NO | CND | NO | CND | **YES** | CND | NO |
| 198-W | CND | NO | CND | NO | CND | **YES** | NO | NO |
| 199-TP | CND | NO | CND | NO | NO | **YES** | CND | NO |
| 200-W | CND | NO | CND | NO | NO | **YES** | CND | NO |
| 201-W | CND | NO | CND | NO | NO | **YES** | NO | NO |
| 202-TP | NO | NO | CND | NO | NO | **YES** | NO | NO |
| 203-S | CND | NO | CND | NO | NO | **YES** | NO | NO |
| 204-S | CND | NO | CND | NO | NO | **YES** | CND | NO |
| 205-S | CND | NO | CND | NO | NO | **YES** | NO | NO |
| 206-S | CND | NO | CND | NO | NO | **YES** | CND | NO |
| 207-W | CND | NO | CND | NO | NO | CND | NO | NO |
| 208-TP | CND | NO | CND | NO | NO | NO | NO | NO |
| 209-TP | CND | NO | CND | NO | CND | **YES** | CND | NO |
| 210-W | CND | NO | CND | NO | NO | **YES** | CND | NO |
| 211-W | CND | NO | CND | NO | CND | **YES** | NO | NO |
| 212-TP | CND | NO | CND | NO | NO | **YES** | NO | NO |
| 213-S | CND | NO | CND | NO | NO | **YES** | CND | NO |
| 214-TP | CND | NO | CND | NO | NO | CND | CND | NO |
| 215-W | CND | NO | CND | NO | NO | CND | NO | NO |

(Continued)

| ID | Cr | Zn | Ni | Pb | Hg | Cu | Cd | As |
|---|---|---|---|---|---|---|---|---|
| 216-S | CND | NO | CND | NO | NO | **YES** | NO | NO |
| 217-TP | NO | NO | CND | NO | NO | CND | NO | NO |
| 218-W | CND | NO | CND | NO | NO | **YES** | NO | NO |
| 219-TP | CND | NO | CND | NO | NO | **YES** | CND | NO |
| 220-W | CND | NO | CND | NO | NO | **YES** | NO | NO |
| 221-S | CND | NO | CND | NO | NO | **YES** | NO | NO |
| 222-S | CND | NO | CND | NO | NO | **YES** | CND | NO |

Notes:
"NO": no intervention, "YES": unconditional intervention, and "CND": conditional intervention.
Each row corresponds to a tailing sample from abandoned deposits, the columns show if this sampling suggests a conditional intervention, an unconditional intervention, or no intervention according to the adjusted threshold factor.

Table 7 shows a summary of results obtained for each sample of the SERNAGEOMIN data for abandoned deposits. It should be noted that in almost all cases it is necessary to carry out an unconditional intervention in each deposit due to the high concentrations of copper. If the particular case of copper which requires unconditional intervention in all tailings is not considered, it can be observed that, although the reasons for conditional intervention might be different in each case, the element nickel in all cases suggests a conditional intervention.

## Weighted intervention ranking

The results show that in the vast majority of the tailings it is necessary to carry out an intervention due to the high concentration of copper. Although there is a great variability between the adjusted threshold factors for the different deposits, the fact that the vast majority of them are above the unconditional intervention limit makes prioritization difficult, even if they were ordered by $AC_F$ results. Given this situation, copper concentration in each tailing and their respective adjusted threshold factor is not a good indicator to provide a prioritization to interventions, due to which it is necessary to be guided by the results of the other elements in this case.

According to the above, it can be seen that for all the other elements analyzed (Cd, Pb, Zn, Cr, As, Ni, and Hg) a significant number of the evaluated sites are in the category of conditional intervention or unconditional intervention. A simple alternative to prioritize the sites that require an expeditious intervention is starting with the sites that have the highest number of elements that require intervention (conditional or unconditional). Nevertheless, a method based on a linear model of weighted costs according to the health and environmental risk represented by each element is proposed.

For these reasons, this work proposes the use of a weighted intervention ranking ($WIR_j$) of the $j$-th site ($1 \leq j \leq m$, where $m$ is the number of sites of the study) and is defined according to Eq. (4).

$$WIR_j = \sum_{i=1}^{n} w_i \cdot x_{ij} \tag{4}$$

**Table 8 Assigned weights to each element for the calculation of the weighted intervention ranking.**

| Element | Assigned weight |
|---|---|
| Cr | 2.0 |
| Zn | 1.0 |
| Ni | 3.0 |
| Pb | 3.0 |
| Hg | 2.0 |
| Cu | 1.0 |
| Cd | 3.0 |
| As | 3.0 |

Note:
Each row in this table shows an element with its respective weight for the calculation of the weighted intervention ranking.

**Table 9 Summary of results for the top 10 critical sites according to their WIR value.**

| ID | Deposit | Status | As | Cd | Pb | Ni | Hg | Cu | Zn | Cr | WIR |
|---|---|---|---|---|---|---|---|---|---|---|---|
| 190-TP-2 | ARIZONA 1 | INACTIVE | 2.45 | 0.84 | 1.21 | 1.27 | −1.73 | 2.02 | 1.26 | 1.39 | 19.88 |
| 190-TP | ARIZONA 1 | INACTIVE | 2.45 | 0.59 | 1.21 | 1.27 | −1.73 | 2.02 | 1.26 | 1.39 | 19.13 |
| 191-W-2 | ARIZONA 2 | INACTIVE | 2.74 | −0.27 | 1.68 | 1.19 | −1.44 | 1.86 | 1.56 | −0.36 | 15.84 |
| 191-W | ARIZONA 2 | INACTIVE | 2.74 | −0.57 | 1.68 | 1.19 | −1.44 | 1.86 | 1.56 | −0.36 | 14.93 |
| 152-TP | SANTA TERESITA 2 | ACTVE | 1.58 | 1.19 | −1.12 | 1.17 | −1.78 | 2.97 | −0.94 | 1.18 | 9.27 |
| 176-TP-2 | ARENILLAS 2 | INACTIVE | −1.33 | 0.84 | −1.67 | 1.23 | 1.73 | 2.97 | −1.18 | 1.14 | 4.71 |
| 176-TP | ARENILLAS 2 | INACTIVE | −1.33 | 0.75 | −1.67 | 1.23 | 1.73 | 2.97 | −1.18 | 1.14 | 4.45 |
| 197-W | PUNTA CALETONES 3 | ABANDONED | −1.33 | 1.62 | −1.62 | 1.16 | 1.82 | 2.61 | −1.00 | −0.22 | 4.30 |
| 209-TP | IRENE 2 | ABANDONED | −1.33 | 1.19 | −1.58 | 1.18 | 0.97 | 2.87 | −0.97 | 0.69 | 3.57 |
| 189-W-2 | ARIZONA 1 | INACTIVE | −1.33 | −1.09 | −1.54 | 1.21 | 2.12 | 2.99 | −0.99 | 1.08 | 3.14 |

Note:
Each row represents a tailing sample, this table details the values of the adjusted threshold factor for each element and the value of the WIR, it also includes the information about the deposit from where the sample was extracted.

Where $n$ stands for the number of elements of interest in the analysis of the tailings (in the case of this article $n = 8$), $1 \leq w_i \leq 5$ is an integer that represents the influence of the $i$-th element on the WIR and $x_{ij}$ corresponds to the adjusted threshold factor for the $i$-th element in the $j$-th site of interest.

The definition of the weights can be controlled by the user of the methodology, who can assign different values to the weights according to environmental, economic and legal criteria. In Table 8 the weighting used in this work is shown. Note that the values can be modified according to the needs of each analysis. Our values are based on the work of *Adriano (2001)*, in particular, we use the toxicity information for plants, animals, and humans through the following criteria: if a metal is potentially toxic to plants, animals, and humans, it is given a weight of 3; if it is only toxic to two of these three, then it is given a weight of 2; similarly, if it is only potentially toxic to one of them, a score of 1 is given. Since it would not make sense to analyze the case of a zero-weight metal in this study, the actual weights range from 1 to 3.

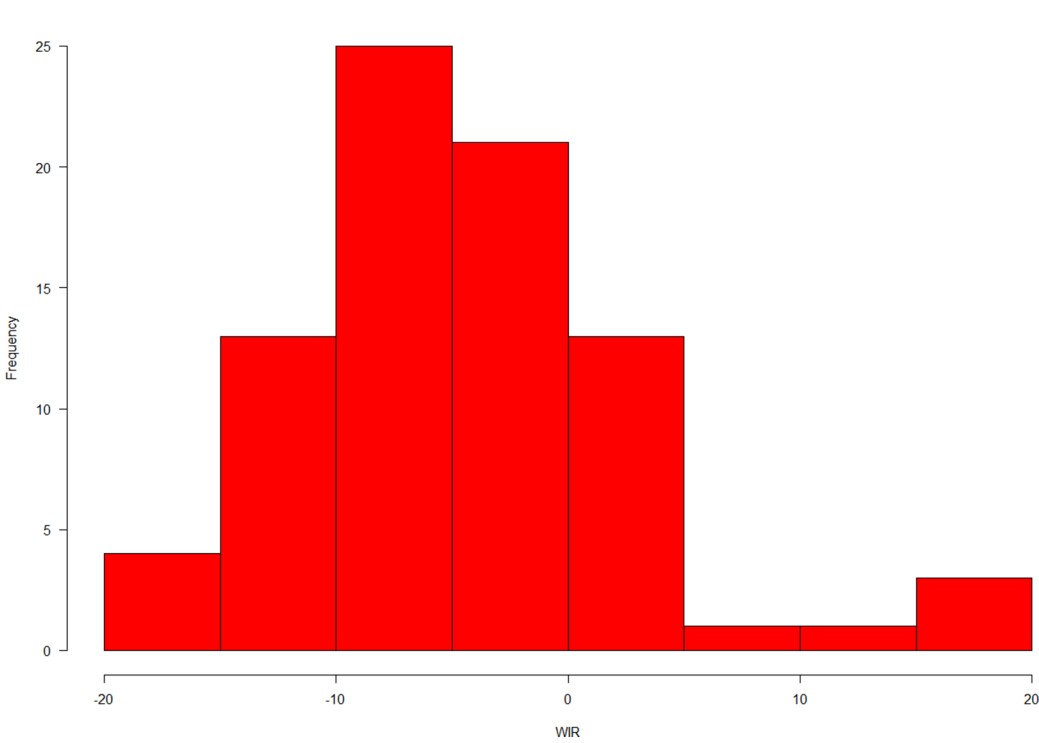

**Figure 7 Histogram for the values of weighted intervention ranking for all the 81 samples from the commune of Andacollo.** This graph shows the distribution of the weighted intervention ranking (WIR) for all the samples from the commune of Andacollo. The average *WIR* for all sites is −4.53, while the median is −5,71 and the standard deviation is 7.53. The highest value of *WIR* is 19.88 and the lowest value is −16.28.

According to the values provided in Table 8, WIRs can be obtained. The results obtained for the six sites with the highest WIRs are shown in Table 9. The average *WIR* for all sites is −4.53, while the median is −5.71 and the standard deviation is 7.53. The highest value of *WIR* is 19.88 and the lowest value is −16.28. Figure 7 shows the distribution of the WIRs.

On the other hand, the difference between the fourth and the fifth sites of highest WIR in Table 9 should be noted. It should also be highlighted that the top four values are more than two standard deviations above average. Considering the above, this point lends itself as a natural limit to define a threshold regarding intervention priority, at least in a first stage.

Based on these results, it is estimated that the first priority of intervention corresponds to the deposits ARIZONA 1 and ARIZONA 2, due to their high *WIR* value. In a subsequent stage, SANTA TERESITA 2 and ARENILLAS 2 should be intervened. It is thus necessary to design an intervention plan for these deposits. On the other hand, if we consider a value of 0 WIR as the cut-off for priority intervention, then 18 out of the 81 (22.22%) samples suggest requirements of intervention according to this metric.

The proposed methodology was applied on 81 samples of available geochemical data (22 tailings deposits), obtaining that 18 samples indicate a critical intervention

requirement (with 10 coming from inactive deposits, one from an active deposit and seven from abandoned deposits). Of these samples, 11 come from tailings associated with Cu–Au extraction processes and seven come from tailings associated with Au extraction processes. Of the corresponding tailings, the one with greater tonnage corresponds to "Punta Caletones 4," which is in a state of abandonment with an admitted capacity of 227,850 tons.

Of course, the exact intervention plan and their feasibility depend on an analysis of environmental impact and economic and legal aspects out of the scope of this work, since the aim of the methodology is to indicate the sites that should be prioritized according to the defined criteria. Having shown the calculation and application of the *WIR*, the exposition of the evaluation methodology proposed in this work is concluded.

As a further analysis of our results, we perform a one-way ANOVA with the different deposit states (Active, Inactive, and Abandoned) as the main factor. Our null hypothesis is that the state of the deposit has no influence on the WIR values, while our alternative hypothesis is that the state of the deposit does have an effect on the WIR values. Our results indicate that there is a significant difference between the value of the WIR depending on the state of the deposit ($p$-value = 0.005). In particular, Tukey's test suggests that there are no significant differences between Active and Abandoned deposits ($p$-value > 0.10), but Inactive deposits are different, showing a higher mean WIR than Active ($p$-value = 0.045) and Abandoned deposits ($p$-value = 0.006).

## Analysis of results

Out of the abandoned tailings in the commune of Andacollo, nine of them are located inside the urban area, in daily coexistence with the population, constituting a potential risk to the health of people (*Morales, 2013*). These are found in the open and can be destabilized both physically and chemically, as a result of the interaction with external factors. For example, it is known that Chile is a highly seismic country, which means that in the face of a large-scale earthquake, houses, and road structures could collapse and be wiped out, causing the death of people. This occurred in Concepción (Chile) during the 2010 earthquake, as liquefied mine tailings hit a farmer's house, burying people alive under the tailing mass (*Verdugo et al., 2010*).

On the other hand, the interaction of sulfide with air and rainfall generates acid mine drainage, which tends to have a great impact on ecosystems, water resources, and soils (*Kefeni, Msagati & Mamba, 2017*). The lost spaces occupied by these tailings should also be taken into account because they cannot be used for other activities such as agriculture, residential development, or others.

Since the tailings in Andacollo are located in the heart of the city, the handling of the toxic metal for any type of intervention becomes very difficult and dangerous, so it requires special treatment. However, there are not many precedents on how to perform this task, since in general the tailings are in remote places of the communities.

The following two aspects must be taken into account when performing tailings intervention. On the one hand, the potential environmental impact, which puts people and the environment at risk; and on the other, an economic one, since current

technology was not available when they were exploited, it is possible that they contain high-grade ore according to current standards; this represents a potential for the recovery of metals with a commercial interest.

When we speak of intervention, we refer to it as an action that avoids the direct contact of the tailings with people, with water systems (surface and groundwater) or with ecosystems. The main objective of the measures is to stabilize the tailings, both physically and chemically, in order to protect the health and safety of people and the environment.

Finally, it is important to point out that although the methodology applied allows ranking the tailings of Andacollo, this would be the first phase in the tailings' assessment, which would allow assigning intervention priorities. The results should be corroborated with a risk assessment, which should consider the size of the tailings, the proximity to potentially impacted receptors (human and ecological receptors) and any other factor considered important.

According to the obtained results, of the 81 samples evaluated, it was found that 18 require a potential intervention, of these samples seven are associated to abandoned tailings, one to active tailings and 10 to inactive tailings. While the assets and liabilities belong to the responsible companies that according to the new Chilean legislation (Law 20,551 on mine closure plans) must take charge of them, ensuring physical and chemical stability in perpetuity, this does not occur with abandoned tailings, on which it is urgent to carry out a more thorough study that allows prioritizing the intervention requirements. The accepted tonnages of the critical abandoned tailings found in this work fluctuate between 5,872 and 227,850 tons. For the purposes of a subsequent study, these data are of great importance, either to solve the problem with compounds of environmental connotation or to recover metals of commercial interest.

In an international context, this work complements the Dutch regulations, exhaustively detailing the definition and application of the method. In addition, an example of its application is given in an area considered critical in the Chilean national environment. The latter is especially important because Chile does not have specialized regulations and the availability of experimental data is deficient, so this methodology is highly appropriate for this context. Given the above, this proposal could be applied in countries with similar conditions, that is, countries that do not have specialized national regulations or high availability of the required data.

At a global level, there are many tailings that are abandoned, many of these are historical tailings and therefore there is a lack of information regarding their characterization. Since some of these tailings might occupy high volumes in addition to their high heterogeneity, their characterization involves high costs and could require extensive financing. Furthermore, given their age, no organization or person can be held responsible for them, as occurs with the abandoned tailings in Chile. Thus, it is the state that must take charge of the situation, but this requires adequate financing. The high costs render the amount of interventions required unfeasible, so it is highly necessary to have a cost-effective tool to rank these tailings, which in turn will lead to prioritization. Based on this, characterizations can be made with an appropriate sampling design, in order to perform the corresponding risk assessment.

## CONCLUSIONS

This article has exposed a detailed methodology to analyze the requirements of soil intervention. This methodology is based on the stringent and thoroughly tested Dutch regulation for soil remediation (2013 version). The main contribution of this work is the definition of the conditional and unconditional intervention thresholds and the simple graphical method. A case study in the commune of Andacollo in Chile has been detailed, the methodology has been applied successfully, revealing several sites that require both unconditional and conditional intervention.

For the threshold values used by this methodology, the classic intervention value formula provided by the Dutch normative has been modified and adapted to provide a simpler calculation approach for mine tailings, where it can safely be assumed that organic matter is negligible. This approach can be adapted to other kinds of soil provided that a similar assumption can be made about their characterization. In particular, the values and formulas provided in this article can be applied to any soil where organic matter can be assumed to be insignificant. Finally, the results obtained in the case study indicate the necessity of intervention of the tailings. Unconditional interventions being more severe and requiring a more immediate attention. On the other hand, conditional intervention might not be necessary depending on the clay percentage of the soil. Thus, a more detailed analysis is required for these tailings.

One of the potential limitations of the proposed methodology and the performed analysis is that the Andacollo background soil values have not been explicitly considered. Adding this information could improve the accuracy of the results. On the other hand, in the case of samples associated with a conditional intervention, it would also be necessary to have the clay percentage in order to obtain a more accurate estimate. However, in the case of unconditional intervention, the methodology would not need this information. Nevertheless, as an initial estimate of the potential environmental risk of these tailings, the methodology delivers appropriate results.

If the deposit concentration delivers an adjusted threshold value greater than zero and lower than the threshold value for unconditional intervention, then it could be considered as slightly contaminated (requiring conditional intervention). If it exceeds the threshold value, it will be necessary to proceed with an intervention strategy. However, this intervention must be carried out case by case; in particular, this analysis must consider all the metals present in the deposit, and this is why the calculation of the WIR is important since it allows assigning weights to the importance of each element with respect to its associated risks. The results obtained with this methodology are only the first step to guide the study. The final decision on an intervention, if it is required, will depend on the evaluation of the risks and environmental conditions of the site.

### Funding

The authors received no funding for this work.

## Competing Interests

The authors declare that they have no competing interests.

## Author Contributions

- Elizabeth Lam Esquenazi analyzed the data, contributed reagents/materials/analysis tools, prepared figures and/or tables, approved the final draft.
- Brian Keith Norambuena analyzed the data, contributed reagents/materials/analysis tools, prepared figures and/or tables, approved the final draft.
- Ítalo Montofré Bacigalupo authored or reviewed drafts of the paper, approved the final draft.
- María Gálvez Estay authored or reviewed drafts of the paper, approved the final draft.

## Data Availability

The data sets used are public and available here: http://sitiohistorico.sernageomin.cl/mineria-relaves.php.

## Supplemental Information

Supplemental information for this article can be found online at http://dx.doi.org/10.7717/peerj.5879#supplemental-information.

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
