# Peer review of "Evaluation of soil intervention values in mine tailings in northern Chile"

_PeerJ, doi:10.7717/peerj.5879_

## Round 0.1 · original submission · Minor Revisions

The manuscript is well written, uses a sound methodology and makes a significant contribution to scientific knowledge and decision-making. Nevertheless, it can be improved taking into account the comments of the three anonymous reviewers.
The weakest point, in my opinion, is the definition of the weighted intervention ranking, particularly the assignment of weights to the different elements, which is not sufficiently justified. An effort should be made to strengthen this part of the work.
Please consider the comments of the reviewers and the annotated file I enclose.

Reviewer 1 ·

Basic reporting

In light of the clear amount of work that has gone into this study, I'm afraid I must highlight some terminal inadequacies.

The English throughout the manuscript is relatively good, however there are sections of poor grammar and instances of duplicated text (e.g. L304-309 in the methods).

The number of tables (17) and figures (10) is extraordinarily high. Most of these should be relegated to appendices, omitted completely, or condensed into one or two much more informative tables/figure. E.g., Figures 1-9 could be presented as a single plate.

Much of the results section belongs in the materials and methods, e.g. everything up to L353.

I cannot find a single statistical test anywhere.

I understand that this is a methodological paper primarily, but there are no hypotheses presented or tested and the results are extremely repetitive with no data presented or analysed outside of tables and figures. Metal by metal might be interesting, but more informative would be a meta analysis of all sites - e.g., of the 696 tailings dams, how many by your measures require intervention?

The first chunk of the discussion (up to L453) is literally just a table by table description of the data, while the following section is more results than discussion (almost entirely results, actually). There is not a single reference in the entire "discussion", and other than stating the method has been successfully applied there is absolutely no discussion of the implications of the requirement for intervention e.g. ecological impacts, human health impacts, costs, abandonment issues.

There is great data and the potential for significant research findings out of this study, but I'm afraid in its current form it is little more than a rambling simplistic overview of a regional intervention assessment with very little significance either to the international community or to tailings management more broadly. There needs to be much, much more critical thought behind this if it's to be published in the international literature.

Experimental design

The methods are described (in great detail), but there is little originality or rigorous experimental practice. There is potential, but little rigorous examination short of fitting the applied model to a dataset and reporting the output.

Validity of the findings

See comments in section 1. In its current form, this paper has very little impact, poor communication of results and conclusions, and is woefully lacking in international context.

Additional comments

This paper doesn't do justice to the data and effort behind applying the model, for which the authors should be commended. But I'm afraid it requires very significant revision (shorter, concise methods; rewritten concise results; competely rewritten discussion that addresses the context of the study; conclusions in an international context; revised tables and figures).

Reviewer 2 ·

Basic reporting

Clear and unambiguous, professional English used throughout.

The paragraphs are too short. Hence, two or more paragraphs must be joined together.
There are repetitions in the texts (e.g. Section 3.1.)

References are alright. Sufficient field background has been provided.

Article structure along with Figure and Tables is ok.

Experimental design

Original primary research within Aims and Scope of the Journal.

Research question well defined, relevant & meaningful.

Validity of the findings

Data is robust, statistically sound, & controlled.

Conclusions are well stated and linked to original research question.

Reviewer 3 ·

Basic reporting

Dear editor

Thank you for the invitation to review "Evaluation of soil intervention values in mine tailings in northern Chile".

Although I request more days to review this paper, is too long to review in 10-20 days. I tried to review it although I do not sure if I did a good review.

In my opinion, the submitted paper has a high value for their publication in Peer J. In general, I have not got a great number of issues about them. I think that should be accepted after minor changes.

I have not opinioned about English quality because I am not English native.

Experimental design

Research is well-designed.

Validity of the findings

Submitted results are very interesting for other researchers.

Additional comments

Abstract

- I suggest a reduction in materials and methods and improve results or conclusions. Authors only wrote about their method but not about their findings.

Introduction

Lines 63-77.

I suggest that authors give some data about mining and their impact on Chilean environment. I know that Cu is the most valuable element in Chilean mining, but it is missing in this section. Also, if possible give an approximate number around mines in Chile. For example, "Chile has more than XXXX mining sites where Cu, Pb and XXX are the most valuable elements".

Please, add a reference for lines 76-77.

Line 89-94. I think that you can reduce some references or merged some of them.


Regarding to material and methods and results sections, I have not several comments about them. They are very well.

In my opinion, figures quality should be improved.

---

## Round 0.2 · Minor Revisions

I am sorry for again recommending minor revisions, but I would like to get the best from your work. You have made a big effort and addressed the comments of all reviewers and also mine. However, I have a few minor comments on the newly added paragraphs (see also the annotated manuscript):

• The names of chemical elements must be written in lowercase (see annotated manuscript).
• Lines 494-500 should go after line 574.
• Line 602: replace “sulfur” by “sulphide” (or “sulfide”)
• See my comment on lines 602-603.
• There is a problem with numbering sections and subsections: Section 3. starts with 3.5. Section 4. starts with 4.5.
• I do not understand that the Discussion starts with “Summary”
• Conclusions are now too long. I recommend moving some information to the Discussion.

Reviewer 3 ·

Basic reporting

I'm not native english speaker and I have not got an opinion about language quality.

Experimental design

No comment

Validity of the findings

No comment

Additional comments

Although it could be very local, I think that it is very interesting for other researchers.

In addition, this paper was improved since the last submission.

Besides, this paper was improved since the last submission.

---

## Round 0.3 · accepted · Accept

You have made a great effort to improve the manuscript. I am happy to accept it for publication in PeerJ.